# Regulatory switch at the cytoplasmic interface controls TRPV channel gating

**Lejla Zubcevic[1][†], William F Borschel[1][†], Allen L Hsu[2], Mario J Borgnia[1,2], Seok-Yong Lee[1]***

[1]Department of Biochemistry, Duke University School of Medicine, Durham, United States; [2]Genome Integrity and Structural Biology Laboratory, Department of Health and Human Services, National Institute of Environmental Health Sciences, National Institutes of Health, Durham, United States

**Abstract** Temperature-sensitive transient receptor potential vanilloid (thermoTRPV) channels are activated by ligands and heat, and are involved in various physiological processes. ThermoTRPV channels possess a large cytoplasmic ring consisting of N-terminal ankyrin repeat domains (ARD) and C-terminal domains (CTD). The cytoplasmic inter-protomer interface is unique and consists of a CTD coiled around a β-sheet which makes contacts with the neighboring ARD. Despite much existing evidence that the cytoplasmic ring is important for thermoTRPV function, the mechanism by which this unique structure is involved in thermoTRPV gating has not been clear. Here, we present cryo-EM and electrophysiological studies which demonstrate that TRPV3 gating involves large rearrangements at the cytoplasmic inter-protomer interface and that this motion triggers coupling between cytoplasmic and transmembrane domains, priming the channel for opening. Furthermore, our studies unveil the role of this interface in the distinct biophysical and physiological properties of individual thermoTRPV subtypes.
DOI: https://doi.org/10.7554/eLife.47746.001

*For correspondence:
seok-yong.lee@duke.edu

[†]These authors contributed equally to this work

**Competing interests:** The authors declare that no competing interests exist.

## Introduction

The Transient Receptor Potential Vanilloid (TRPV) channel subfamily is a subset of the large TRP channel superfamily, and consist of subtypes TRPV1-TRPV6 (*Gunthorpe et al., 2002*). TRPV1-4 are intrinsically temperature sensitive (thermoTRPV) (*Caterina et al., 1997*; *Caterina et al., 1999*; *Güler et al., 2002*; *Smith et al., 2002*) and have been found to play important roles in numerous physiological processes including thermosensation (*Moqrich et al., 2005*; *Marics et al., 2014*; *Patapoutian, 2005*; *Gavva et al., 2008*), nociception (*Caterina et al., 2000*; *Gopinath et al., 2005*; *Bang et al., 2010*; *Reilly and Kym, 2011*; *Huang and Chung, 2013*), and osmosensation (*Zanou et al., 2015*). Recent studies have made great strides in elucidating the molecular mechanisms of ligand-dependent gating and activation of thermoTRPV channels (*Hu et al., 2009*; *Cao et al., 2013*; *Zhang et al., 2016*; *Zubcevic et al., 2018a*; *Zubcevic et al., 2018b*; *Singh et al., 2018*; *Zhang et al., 2019*). However, despite this wealth of structural information, the role of their cytoplasmic domains, which make up the majority of the structure, remains unclear. The importance of these domains in channel gating has long been acknowledged with a number of studies finding that mutations in the cytoplasmic regions can profoundly affect the function of thermoTRPV channels (*Lishko et al., 2007*; *Phelps et al., 2010*; *Shi et al., 2013*; *Landouré et al., 2010*; *Salazar et al., 2008*; *Yao et al., 2011*; *Brauchi et al., 2006*). Cryo-electron microscopy (cryo-EM) and X-ray crystallography studies (*Cao et al., 2013*; *Zubcevic et al., 2018a*; *Zubcevic et al., 2018b*; *Singh et al., 2018*; *Liao et al., 2013*; *Zubcevic et al., 2016*; *Huynh et al., 2016*; *Deng et al., 2018*) have revealed that the cytoplasmic domains of thermoTRPVs are composed of the ankyrin repeat domain (ARD), containing six ankyrin repeats, a coupling domain (CD) that is made up of a β-sheet (β$_{CD}$), a

helix-loop-helix motif ($HLH_{CD}$) and the pre-S1(pre-$S1_{CD}$) helix, and a C-terminal domain (CTD) which extends from the conserved amphipathic helix, termed the TRP domain, into the cytosol where it forms a hair-pin structure and doubles back to the $\beta_{CD}$ to which it contributes a beta strand (*Figure 1A*). Interestingly, a recent study of the human TRPV3 (hTRPV3) channel (*Zubcevic et al., 2018b*) resolved the structure of the CTD beyond the $\beta_{CD}$ and showed that this distal region of the CTD (distal CTD) coils around $\beta_{CD}$ and emerges at the front side of the channel. This coil forms extensive interactions with the ARD of the neighboring protomer and therefore contributes substantially to the cytoplasmic inter-protomer interface. All thermoTRPV channels possess cytoplasmic inter-protomer contacts which are formed by the ARD of one and the CD of the neighboring protomer and the distal CTD region is conserved in these channels. However, a distal CTD coil similar to the one observed in TRPV3 was only seen in TRPV2 (*Zubcevic et al., 2018a*) since this region was poorly resolved in the structures of TRPV1 and TRPV4 (*Cao et al., 2013*; *Liao et al., 2013*; *Deng et al., 2018*).

In order to dissect the role of this unique cytoplasmic interface in thermoTRPV channel gating, we have conducted structural and functional studies using human TRPV3 as a model system. TRPV3 exhibits use-dependent increase in current amplitudes (termed sensitization) upon repeated applications of either ligand or heat. This phenomenon arises from hysteresis (irreversible change) of TRPV3 gating and can be adequately described by a simple gating scheme with one open and two closed states (*Liu et al., 2011*) (*Figure 1—figure supplement 1*). Following stimulation, the return rate to the initial unliganded closed state ($C_0$) of the channel is slow enough to be irreversible. Instead, the channels transition to either an intermediate (sensitized) closed state ($C_1$ in Scheme 1, *Figure 1—figure supplement 1*) or a new sensitized resting state after opening ($C_1$ in Scheme 2, *Figure 1—figure supplement 1*). Repeated stimulation leads to an increase in $C_1$ occupancy reflected in progressive increase in current in response to stimuli. Therefore, the $C_1$ state represents a closed state with a lower energy requirement for opening. Sensitization properties of thermoTRPVs are strongly subtype dependent: while TRPV3 (*Liu et al., 2011*) and TRPV2 (*Liu and Qin, 2016*) both sensitize upon repeated stimulation with heat or agonists, analogous sensitization upon stimulation with capsaicin or heat has not been observed in TRPV1, suggesting that TRPV1 channels do not undergo an irreversible conformational change following activation by a stimuli, and might populate a closed 'sensitized' $C_1$ state from the outset (*Liu and Qin, 2016*).

Our structural and functional studies show that the cytoplasmic assembly plays a critical role in activation of thermoTRPV channels through large structural rearrangements at the CTD-ARD interface that involve secondary structure transitions. The data also provides an explanation for the phenotypic differences amongst the thermoTRPV channels and sheds light on the role of the cytoplasmic inter-protomer interface in thermoTRPV gating.

## Results

### Disrupting the interactions between CTD and ARD results in sensitized channels

Our previously determined structures of the hTRPV3 channel revealed that the distal CTD coils around $\beta_{CD}$ and establishes a number of interactions with the ARD of the neighboring protomer (*Zubcevic et al., 2018b*). Notably, the acidic residues E751 and D752 in the distal CTD form a salt-bridge with the K169 residue on the neighboring ARD (*Figure 1B*). A previous study reported that mutation of K169A altered sensitivity to ligands and proposed allosteric modulators of TRPV3 channel function; however, the underlying mechanism was elusive as the study was conducted prior to structural elucidation of the full-length TRPV channels (*Phelps et al., 2010*). With the knowledge that K169 forms a part of the CTD-ARD interface, we set out to investigate its role in channel gating. Wild-type hTRPV3 channels show a steady use-dependent increase in the current response upon successive applications of 30 μM 2-Aminoethoxydiphenyl borate (2-APB) for ~30–40 cycles of stimulation before reaching saturation (*Figure 1C–D*, *Figure 1—source data 1*). By contrast, the K169A mutant appears to be fully sensitized, as the first few applications of ligand typically elicit the maximal current response which does not increase upon further stimulation, suggesting the channel is sensitized (*Figure 1E–F*, *Figure 1—source data 1*). In order to determine if this sensitized phenotype is the result of disruption of the salt-bridge interaction between ARD and CTD, we introduced

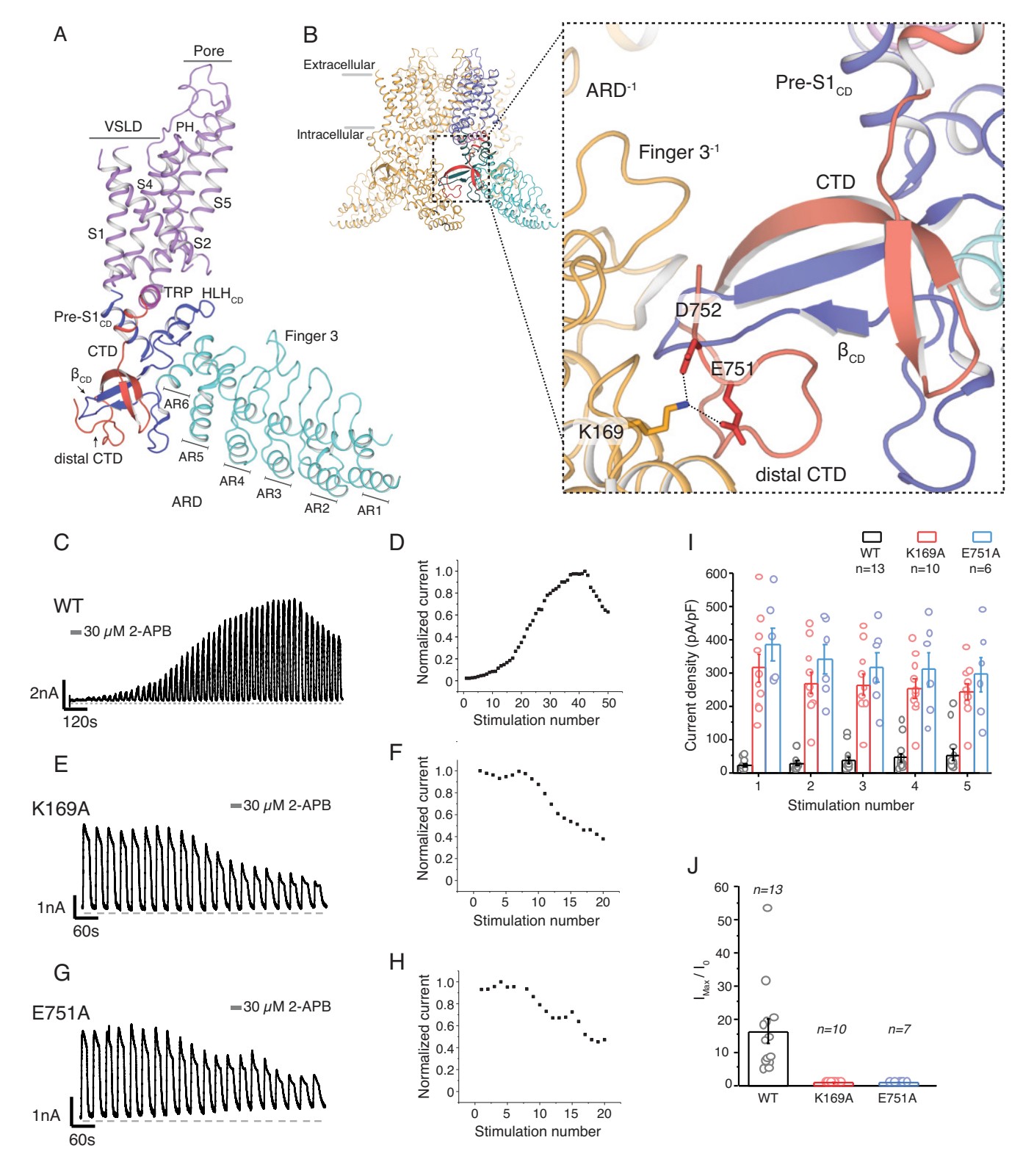

**Figure 1.** The role of the cytoplasmic inter-protomer interface in hTRPV3 gating. (A) Architecture of the hTRPV3 protomer. Ankyrin repeat domain (ARD) is colored in cyan, the coupling domain (CD) and its individual elements (HLH_CD, β_CD, Pre-S1_CD) are colored in blue, transmembrane helices S1-S6 are colored in violet, the TRP domain is shown in magenta and the C-terminal domain (CTD) is colored in red. (B) A close-up view of the inter-protomer interface in hTRPV3. Residue K169 from the ARD and residues E751 and D752 from the CTD are shown in stick representation. Representative

*Figure 1 continued on next page*

*Figure 1 continued*

whole-cell current traces recorded at +60 mV from WT (**C–D**), K169A (**E–F**), and E751A (**G–H**) evoked by repeating applications 30 µM 2-APB for 15 s followed by 15 s of washout and corresponding time-course of use-dependent changes in the relative current amplitude. (**I**) Average current density from the first five 2-APB stimulations (WT: n = 13 biologically independent experiments; K169A: n = 10 biologically independent experiments; E751A: n = 6 biologically independent experiments). (**J**) Initial sensitization was characterized by the ratio of the response to 2-APB during the first ($I_0$) and maximum current ($I_{max}$) response ($I_{max}/I_0$) calculated as the mean from each biologically independent experiment. (WT: n = 13 biologically independent experiments; K169A: n = 10 biologically independent experiments; E751A: n = 7 biologically independent experiments).

DOI: https://doi.org/10.7554/eLife.47746.002

The following source data and figure supplement are available for figure 1:

**Source data 1.** This spreadsheet contains data points for the time course of 2-APB use-dependence plots in *Figure 1D, 1F and 1H*; current density data used to generate bar plots in *Figure 1I*; initial sensitization data used to generate bar plots in *Figure 1J*.

DOI: https://doi.org/10.7554/eLife.47746.004

**Figure supplement 1.** State model and gating schemes of TRPV3 channels.

DOI: https://doi.org/10.7554/eLife.47746.003

a E751A mutation in the CTD of the hTRPV3 channel. Indeed, similar to K169A, neutralizing the acidic E751 residue produced a sensitized phenotype, characterized by large saturating currents upon first application of ligand (*Figure 1G–H*, *Figure 1—source data 1*). Together these data suggest that the salt-bridge formed by K169 and E751 at the CTD-ARD interface plays a critical role in hysteresis and that breaking these interactions changes the occupancy of the $C_0$ and $C_1$ states before stimulation, resulting in sensitized hTRPV3 channels (*Figure 1I–J*, *Figure 1—source data 1*).

## Structure of the K169A mutant reveals large conformational changes in the distal CTD

In order to elucidate the conformational changes that underlie the transition to the sensitized $C_1$ state, we introduced the K169A mutation into the hTRPV3 carrying the previously reported functionally silent T96A mutation (*Zubcevic et al., 2018b*) (TRPV3$_{K169A}$ in future references). We expressed and purified the TRPV3$_{K169A}$ channel and solved its structure by single particle 3D cryo-electron microscopy (cryo-EM) (*Table 1*, *Figure 2—figure supplements 1* and *2*). Remarkably, the cytoplasmic assembly of TRPV3$_{K169A}$ undergoes large structural rearrangements while the overall conformation of the transmembrane domains (TM) of the TRPV3$_{K169A}$ channel closely resembles that of the previously reported hTRPV3 in its closed, apo form (PDB ID 6MHO, TRPV3$_{WT}$) (*Zubcevic et al., 2018b*) (*Figure 2A*). The most drastic change occurs at the CTD. In the TRPV3$_{WT}$, the distal CTD coils around β$_{CD}$ and establishes a large interface between the CD and the ARD of neighboring subunits. By contrast, the K169A mutation induces a substantial change in the secondary structure and the position of the distal CTD as well as the ARD-CTD interface. Specifically, the distal CTD undergoes a dramatic coil-to-helix transition and this newly formed helical distal CTD is positioned behind the β$_{CD}$ in the cytoplasmic vestibule of the channel (*Figure 2B*, *Figure 2—figure supplement 2*). Furthermore, the cryo-EM map of the TRPV3$_{K169A}$ contains a protein density that abuts the proximal CTD and the CD and occupies a similar space to that vacated by the distal CTD upon coil-to-helix transition. It is unlikely that this density forms a part of the CTD as the cryo-EM map shows no connectivity to the CTD helix. Instead, the density appears to be connected to the N-terminal of the neighboring protomer to which connectivity is visible at low contours of the map (*Figure 2B*, *Figure 2—figure supplement 2*). Since this density is not sufficiently resolved to allow for unambiguous model building, we built this putative N-terminal region as a polyalanine chain.

## Conformational changes of the distal CTD lead to rearrangements in the cytoplasmic inter-subunit interface

The coil-to-helix transition in the distal CTD and the rearrangement of the putative N-terminal region in the TRPV3$_{K169A}$ are accompanied by an apparent anti-clockwise rotation of the ARD when viewed from the extracellular space (*Figure 3—figure supplement 1*). This rotation does not reflect a strict rigid body movement of the ARD as the tetrameric assembly of TRPV3$_{K169A}$ cytoplasmic domains cannot be superposed well with that of TRPV3$_{WT}$ through mere rotation (Cα R.M.S.D. 2.3 Å) (*Figure 3—figure supplement 1*). Nevertheless, individual ARDs from the two structures superpose well (Cα R.M.S.D. 0.9 Å) (*Figure 3—figure supplement 1*), revealing that the ARDs of each TRPV3$_{K169A}$

**Table 1.** Cryo-EM data collection, refinement and validation statistics

| | TRPV3$_{K169A}$ (EMD-20192) (PDB 6OT2) | TRPV3$_{K169A\ 2-APB}$ (EMD-20194) (PDB 6OT5) |
|---|---|---|
| Data collection and processing | | |
| Magnification | 130,000x | 75,000x |
| Voltage (kV) | 300 | 300 |
| Electron exposure (e–/Å$^2$) | 40 | 42 |
| Defocus range (μm) | 1–2.5 | 1.25–3 |
| Pixel size (Å) | 1.06 | 1.08 |
| Symmetry imposed | C4 | C4 |
| Initial particle images (no.) | 452,388 | 1,174,521 |
| Final particle images (no.) | 95,184 | 79,006 |
| Map resolution (Å) | 4.1 | 3.6 |
| FSC threshold | 0.143 | 0.143 |
| Refinement | | |
| Initial model used (PDB code) | 6MHO | 6MHO |
| Model resolution (Å) | 4.1 | 3.6 |
| FSC threshold | 0.143 | 0.143 |
| Map sharpening B factor (Å$^2$) | −120 | −100 |
| Model composition | | |
| Non-hydrogen atoms | 17,332 | 17,800 |
| Protein residues | 2500 | 2492 |
| Ligands | 0 | 4 (2-APB) |
| B factors (Å$^2$) | | |
| Protein | 87.43 | 40.51 |
| Ligand | n/a | 35.66 |
| R.m.s. deviations | | |
| Bond lengths (Å) | 0.008 | 0.008 |
| Bond angles (°) | 0.868 | 0.833 |
| MolProbity score | 1.64 | 1.24 |
| Clashscore | 5 | 5 |
| Poor rotamers (%) | 0 | 0 |
| Ramachandran plot | | |
| Favored (%) | 92.70 | 97.01 |
| Allowed (%) | 7.30 | 2.99 |
| Disallowed (%) | 0 | 0 |

DOI: https://doi.org/10.7554/eLife.47746.008

protomer swivel in a manner which lifts the N-terminal part of the ARD towards the membrane while the C-terminal part of the ARD along with the CTD is lowered further into the cytosol (*Figure 3—figure supplement 1*). Furthermore, the coil-to-helix transition also results in extensive changes in the interface between ARD and CTD (*Figure 3—figure supplement 1*), causing a conformational change of the loop of ankyrin repeat 5 (AR5) (*Figure 3A* and *Figure 3—figure supplement 1*). The combined effect of these rearrangements increases the coupling between the CTD and the ARD, as well as between the ARD, the CD and the TRP domain (*Figure 3—figure supplement 1*).

In the TRPV3$_{WT}$ structure, the distal CTD makes contacts with ankyrin repeat 2 (AR2) and the loop connecting AR3 and AR4, termed finger 3, at the front side of the interface (*Figure 1B*) and the loop of AR5 in the intracellular vestibule (the back side of the interface) (*Figure 3A and C*). By contrast, in

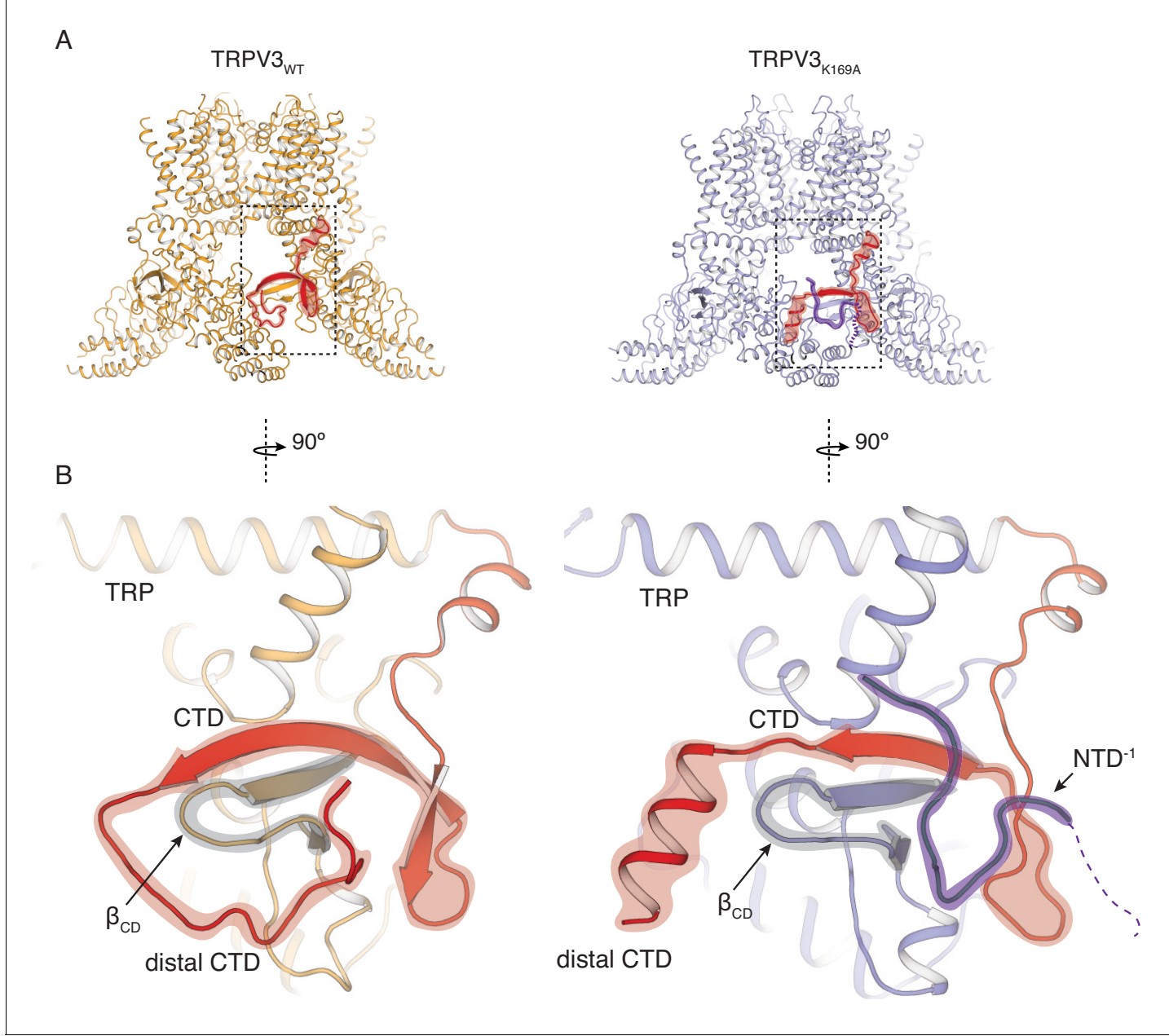

**Figure 2.** Rearrangements of the cytoplasmic domains in the TRPV3$_{K169A}$ structure. (A) The cytoplasmic inter-protomer interface in TRPV3$_{WT}$ (left panel) and TRPV3$_{K169A}$ (right panel). The CTD and the putative N-terminal region are highlighted in red and purple, respectively. (B) Close-up view of the rearrangements in the cytoplasmic domains. In the TRPV3$_{WT}$, the distal CTD (highlighted in red) coils around the β$_{CD}$ (highlighted in grey) (left panel). In the TRPV3$_{K169A}$ structure, the distal CTD undergoes a coil-to-helix transition (highlighted in red). An additional polypeptide density (highlighted in purple) is observed near the front of the β$_{CD}$ (highlighted in grey) and the proximal CTD, in the vicinity of the space occupied by the distal CTD coil in TRPV3$_{WT}$ and was assigned as a putative N-terminal domain from the neighboring protomer (NTD$^{-1}$).

DOI: https://doi.org/10.7554/eLife.47746.005

The following figure supplements are available for figure 2:

**Figure supplement 1.** Cryo-EM data collection and processing, TRPV3$_{K169A}$.
DOI: https://doi.org/10.7554/eLife.47746.006
**Figure supplement 2.** Electron density in the TRPV3$_{K169A}$ cryo-EM map.
DOI: https://doi.org/10.7554/eLife.47746.007

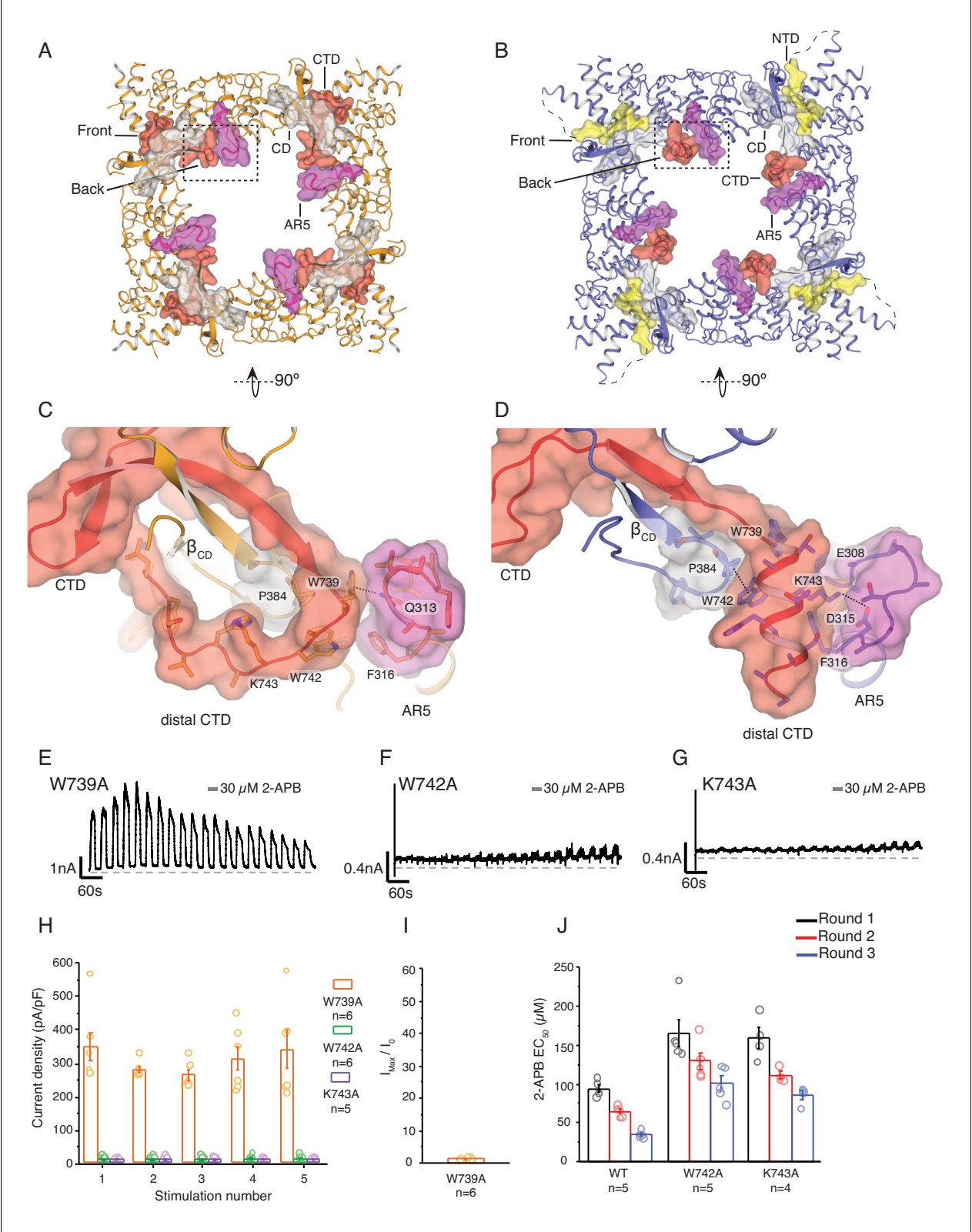

**Figure 3.** State-dependent changes at the cytoplasmic inter-protomer interface. (**A–B**) Top view of the cytoplasmic inter-protomer interactions in TRPV3$_{WT}$ (**A**) and TRPV3$_{K169A}$ (**B**). In TRPV3$_{WT}$ the CTD (red) coils around the β$_{CD}$ (grey). The distal CTD interacts with the ARD at the front of the interface and with the loop of ankyrin repeat 5 (AR5, magenta) at the back. In TRPV3$_{K169A}$, the interface is changed due to the coil-to-helix transition in the distal CTD, which no longer participates in the interactions at the front of the interface and forms tighter interactions with AR5. The front of the

*Figure 3 continued on next page*

*Figure 3 continued*

interface is now occupied by the putative NTD (yellow). (**C**) A close-up view from the cytoplasmic cavity of the interactions between the distal CTD (red surface representation) and AR5 (magenta surface representation) in TRPV3$_{WT}$. Residue W739 forms a cation-$\pi$ interaction with the amino group of Q313 (dashed line). (**D**) The coil-to-helix transition changes the conformation of the AR5 loop. In TRPV3$_{K169A}$, the W739-Q313 interaction is broken. Residues K742 and W743, which in TRPV3$_{WT}$ are not within interaction distances with the rest of the protein, form interactions with the backbone of E308 in AR5 and P384 in $\beta_{CD}$, respectively (dashed lines). Representative whole-cell current traces recorded at +60 mV from W739A (**E**), W742A (**F**), and K743A (**G**) evoked by repeating applications 30 µM 2-APB for 15 s followed by 15 s of washout. (**H**) Average current density for the first five 2-APB stimulations (W739A: n = 6 biologically independent experiments; W742A: n = 6 biologically independent experiments; K743A: n = 5 biologically independent experiments). (**I**) Ratio of first ($I_0$) and maximum current ($I_{max}$) 2-APB stimulation ($I_{max}/I_0$) as in (**G**), calculated as the mean from each biologically independent experiment (W739A: n = 6 biologically independent experiments). (**J**) Mean 2-APB EC$_{50}$ from three consecutive dose-response rounds fit with the Hill equation (WT: n = 5 biologically independent experiments; W742A: n = 5 biologically independent experiments; K743A: n = 4 biologically independent experiments). See *Figure 3—figure supplement 2* for representative current traces and dose–response relationship fit with the Hill equation.

DOI: https://doi.org/10.7554/eLife.47746.009

The following source data and figure supplements are available for figure 3:

**Source data 1.** This spreadsheet contains current density data used to generate bar plots in *Figure 3H*; initial sensitization data used to generate bar plots in *Figure 3I*; calculated dose-response values used to generate bar plots in *Figure 3J*.

DOI: https://doi.org/10.7554/eLife.47746.013

**Figure supplement 1.** Conformational changes in the cytoplasmic domains in TRPV3$_{K169A}$.

DOI: https://doi.org/10.7554/eLife.47746.010

**Figure supplement 2.** The TRPV3 W742A and K743A CTD mutants alter hysteresis.

DOI: https://doi.org/10.7554/eLife.47746.011

**Figure supplement 2—source data 1.** This spreadsheet contains dose-response data used to calculate mean values plotted in *Figure 3—figure supplement 2D*.

DOI: https://doi.org/10.7554/eLife.47746.012

the TRPV3$_{K169A}$ the front side of the interface appears to be formed by the N-terminal region, which extends from the neighboring ARD and abuts the CD and the proximal CTD (*Figures 2B* and *3B*). Notably, while the distal CTD of TRPV3$_{K169A}$ does not contribute to the front side of the interface with minimal interaction with the CD of its own protomer, it forms extensive interactions with the loop of AR5 of the neighboring subunit at the back side of the interface (*Figure 3B and D*). In order to probe the role of the dynamic interactions between the ARD and CTD observed in TRPV3$_{K169A}$, we introduced mutations in the distal CTD and examined their effects on channel function. We chose three sites W739, W742, and K743 for mutational studies. In TRPV3$_{WT}$, W739 forms a cation-$\pi$ interaction with the amino group of Q313 in the loop of AR5 (*Figure 3C*), and this interaction is broken in the TRPV3$_{K169A}$ (*Figure 3D*). Similarly, both W742 and K743 undergo state-dependent changes in their interactions: when the distal CTD adopts a coil conformation W742 and K743 do not form substantial interactions with the rest of the channel (*Figure 3C*), but when the distal CTD is helical W742 forms a CH-$\pi$ interaction with P384 in the CD and K743 is within interaction distance of E308 and D315 in the loop of AR5 (*Figure 3D*). The W739A mutation resulted in a phenotype similar to the sensitized K169A and E751A mutants (*Figure 3E*). By contrast, W742A (*Figure 3F*) and K743A (*Figure 3G*) both resulted in channels with markedly lower activity and decreased ability to sensitize upon repeated stimulation with 30 µM 2-APB (*Figure 3H*, *Figure 3—source data 1*). This reduced activity was not the result of decreased surface expression or non-functional channels, as application of 2-APB at high concentrations (300 µM) resulted in robust current responses (*Figure 3—figure supplement 2*). To further probe this low activity phenotype of W742A and K743A, we examined the effects of these mutations on hysteresis. As the wild-type TRPV3 undergoes hysteresis, its sensitivity to ligand increases, which can be monitored by the reduction of the EC$_{50}$ value (*Figure 3—figure supplement 2*). The initial 2-APB EC$_{50}$ values for both W742A and K743A were higher than that for the wild-type TRPV3 channel (*Figure 3J*; *Figure 3—figure supplement 2*, *Figure 3—source data 1*, *Figure 3—figure supplement 2—source data 1*), but they decreased following each consecutive dose-response round. However, because the 2-APB dose-response curves for W742A and K743A mutants do not reach saturation, the efficacy of 2-APB is likely to be overestimated by these EC$_{50}$ calculations. These results indicate that W742A and K743A are initially less active due to impaired sensitization and might have an increased energy barrier for transitioning to the sensitized conformation. Combined with the results from E751A and K169A,

these data show that mutations to different parts of the distal CTD have distinct effects on sensitization and activation: mutations that destabilize the coil conformation of the distal CTD result in a sensitized phenotype, while mutations that destabilize the helical CTD result in a decrease of hysteresis and sensitization. Taken together, our data suggest that the coil-to-helix transition in the distal CTD and the resulting state-dependent inter-protomer interaction networks are critical for channel gating and that the secondary structure transition in the distal CTD may serve as a switch that controls the entry into the sensitized $C_1$ state.

## Application of 2-APB induces changes in both transmembrane and cytoplasmic assemblies

In order to further investigate the role of the cytoplasmic assembly in activation of the TRPV3 channel, we determined the cryo-EM structure of the TRPV3$_{K169A}$ channel in the presence of 2-APB (TRPV3$_{K169A\ 2-APB}$) to 3.6 Å resolution (*Figure 4—figure supplements 1* and *2*). Inspection of this structure revealed that TRPV3 undergoes conformational changes in both the transmembrane (TM) and cytoplasmic domains in the presence of ligand. The cryo-EM map of TRPV3$_{K169A\ 2-APB}$ contains non-protein densities between the TRP domain and the Pre-S1$_{CD}$ in all four protomers. Because a previous high-throughput mutagenesis study (*Hu et al., 2009*) identified this site as critical for 2-APB binding, we assigned the non-protein densities as 2-APB (*Figure 4A*, *Figure 4—figure supplement 2*). The 2-APB molecule is nestled amongst residues W692, R693 and R696 in the TRP domain and H417, H426, H430 and W433 in the Pre-S1$_{CD}$.

The pore of TRPV3$_{K169A\ 2-APB}$ adopts a putative open conformation similar to that of the recently reported mouse TRPV3 (mTRPV3, PDB ID 6DVZ)) open structure (*Figure 4—figure supplement 3*) and possesses a π-helical turn in the pore-lining S6 helix, which is not present in any structures of the TRPV3 channel that had not been exposed to ligand (*Zubcevic et al., 2018b*; *Singh et al., 2018*), including TRPV3$_{K169A}$ (*Figure 4B* and *Figure 4—figure supplement 2*). Notably, a comparison of S6 helices from the closed TRPV3$_{WT}$ and TRPV3$_{K169A\ 2-APB}$ shows that the linker region between the S6 helix and the TRP domain also changes: while the two helices are connected via a loose loop in the TRPV3$_{WT}$ structure, they form a single helical structure in TRPV3$_{K169A\ 2-APB}$, suggestive of increased coupling between the TRP domain and S6 upon 2-APB binding (*Figure 4C*). In addition, the TRP domain of TRPV3$_{K169A\ 2-APB}$ undergoes a swivel relative to the S6 helix (*Figure 4C*). A closer inspection revealed that the observed swivel in the TRP domain is the result of state-dependent changes in coupling between the transmembrane and the cytoplasmic domains. Namely, the swivel in the TRP domain can be traced back to changes in coupling between the loop of AR5, the HLH$_{CD}$ and the TRP domain which are induced by the coil-to-helix transition in the distal CTD (*Figure 4D–E*). In TRPV3$_{WT}$ the loop of AR5 does not interact with HLH$_{CD}$ (*Figure 4D* and *Figure 4—figure supplement 4*). However, in TRPV3$_{K169A\ 2-APB}$, where the coil-to-helix transition in the distal CTD has enforced a conformational change in the loop of AR5, AR5 interacts with the HLH$_{CD}$. The loop of AR5 pushes on the HLH$_{CD}$, causing a swivel in both the HLH$_{CD}$ and Pre-S1$_{CD}$, leading in turn to a swivel in the TRP domain (*Figure 4E* and *Figure 4—figure supplement 4*). Notably, 2-APB binding between the Pre-S1$_{CD}$ and the TRP domain appears to further increase the coupling between the CD and the TRP domain, suggesting a mechanism for 2-APB-dependent activation of TRPV3 (*Figure 4—figure supplement 4*). In line with our observations, a previous study has shown that manipulation of the length of the loop of HLH$_{CD}$ affects sensitization properties of TRPV3 likely by increasing the coupling between the cytoplasmic and transmembrane domains (*Liu and Qin, 2017*).

Our studies suggest that the distal CTD in TRPV3 plays a critical role in hysteresis, sensitization, and activation. In potential conflict with our findings, the previously reported open structural model of the mutant mouse TRPV3 (mTRPV3) in complex with 2-APB (PDB ID 6DVZ) contains a CTD in the loop conformation (*Figure 4—figure supplement 5*) (*Singh et al., 2018*). However, a closer inspection of the accompanying cryo-EM map (EMD-8921) revealed that this distal CTD density, assigned as a loop in the original study, could more feasibly be built as a helix (calculated correlation around mean equals 0.33 for the loop, and 0.49 for the helix, see Materials and methods, *Figure 4—figure supplement 5*). This further supports our finding that the coil-to-helix transition in the distal CTD is critical for channel activation.

Furthermore, the mTRPV3 study suggested three binding sites for 2-APB: (1) between the TRP domain and Pre-S1$_{CD}$, (2) in the VSLD cavity and (3) at the extracellular interface between helices S1 and S3 of the VSLD (*Singh et al., 2018*). Even though we observe a non-protein density in the VSLD

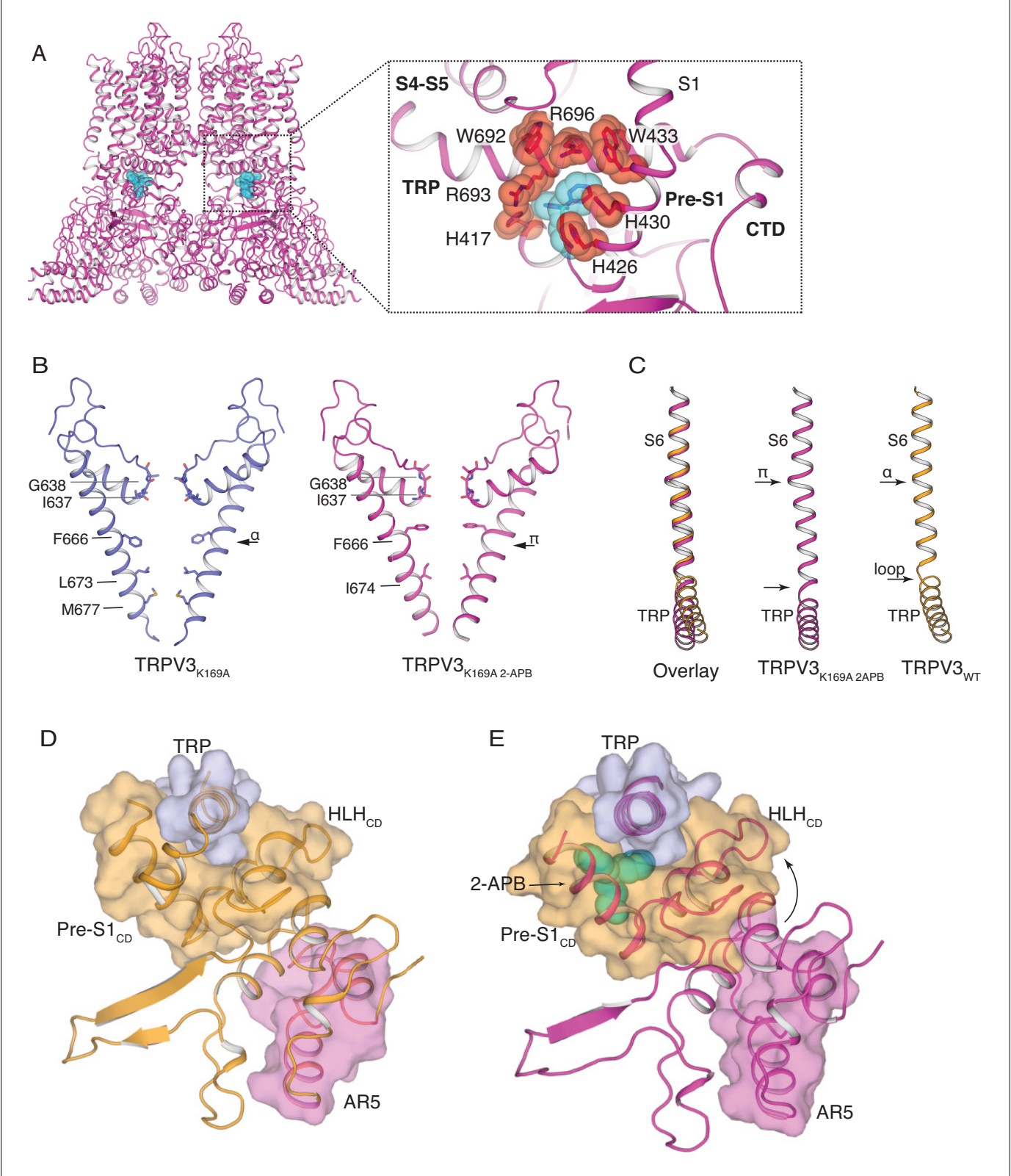

**Figure 4.** The structure of TRPV3$_{K169A\ 2\text{-}APB}$ exhibits changes in both transmembrane and cytoplasmic domains. (**A**) One 2-APB molecule is bound to each protomer of the TRPV3$_{K169A\ 2\text{-}APB}$ channel (magenta). 2-APB is found between the HLH$_{CD}$, Pre-S1$_{CD}$ and TRP domains in a binding site defined by residues H417 in the HLH$_{CD}$, H426, H430, W433 in the Pre-S1$_{CD}$ and W692, R693 and R696 in the TRP domain. All residues are shown in stick and red sphere representation. 2-APB is shown in stick and cyan sphere representation. (**B**) The S6 helix of TRPV3$_{K169A}$ (blue) undergoes an α-to-π transition in

*Figure 4 continued on next page*

*Figure 4 continued*

the presence of 2-APB (magenta). (**C**) The α-to-π transition tightens the connection between S6 and the TRP domain. In the TRPV3$_{WT}$ structure (orange), the TRP domain and the S6 are connected via a loop, but in the TRPV3$_{K169A\ 2-APB}$ channel (magenta) the TRP domain and S6 form a continuous helical structure. In addition, the TRP domain exhibits a swivel in the TRPV3$_{K169A\ 2-APB}$ structure. (**D–E**) The coil-to-helix transition in TRPV3$_{K169A\ 2-APB}$ increases coupling between the cytoplasmic domains and the TRP domain. In the TRPV3$_{WT}$ structure (orange) (**D**), the loop of AR5 (magenta surface) does not interact with the HLH$_{CD}$ (orange surface). However, in TRPV3$_{K169A\ 2-APB}$ (magenta) (**E**) the coil-to-helix transition in the distal CTD induces a conformational change in the loop of AR5 (magenta surface), coupling it to the HLH$_{CD}$ (orange surface) and the TRP domain (light blue surface). 2-APB (cyan stick and surface representation) contributes to increased interactions between the TRP domain and Pre-S1$_{CD}$.
DOI: https://doi.org/10.7554/eLife.47746.014

The following source data and figure supplements are available for figure 4:

**Figure supplement 1.** Cryo-EM data collection and processing, TRPV3$_{K169A\ 2-APB}$.
DOI: https://doi.org/10.7554/eLife.47746.015

**Figure supplement 2.** Electron density in the TRPV3$_{K169A\ 2-APB}$ cryo-EM map.
DOI: https://doi.org/10.7554/eLife.47746.016

**Figure supplement 3.** Comparison of the pore conformations of hTRPV3$_{WT}$, hTRPV3$_{K169A\ 2-APB}$ and mTRPV3$_{Open}$.
DOI: https://doi.org/10.7554/eLife.47746.017

**Figure supplement 4.** Coupling between the cytoplasmic and transmembrane domains.
DOI: https://doi.org/10.7554/eLife.47746.018

**Figure supplement 5.** The CTD in thermoTRPV structures.
DOI: https://doi.org/10.7554/eLife.47746.019

**Figure supplement 6.** APB binding in TRPV3$_{K169A\ 2APB}$.
DOI: https://doi.org/10.7554/eLife.47746.020

**Figure supplement 7.** APB and camphor response of proposed 2-APB binding site mutants.
DOI: https://doi.org/10.7554/eLife.47746.021

**Figure supplement 7—source data 1.** This spreadsheet contains data used to calculate the mean 2-APB to camphor ratio values plotted in *Figure 4—figure supplement 7A*.
DOI: https://doi.org/10.7554/eLife.47746.022

cavity of TRPV3$_{K169A\ 2-APB}$, we did not assign this to 2-APB because our reconstruction of TRPV3$_{K169A}$ also contains a similarly shaped density in this position (*Figure 4—figure supplement 6*). Furthermore, we do not observe a discernible density in the third site proposed by the mTRPV3 study (*Figure 4—figure supplement 6*). Finally, electrophysiological measurements indicate that mutations in hTRPV3 at the proposed second and third sites do not affect the relative responses of 2-APB compared to camphor. By contrast, mutating residues in the first site (H426A) reduces the channels' response to high concentrations of 2-APB without affecting the response to camphor (*Figure 4—figure supplement 7*, *Figure 4—figure supplement 7—source data 1*). This suggests that sites 2 and 3 are not involved in 2-APB-dependent activation of hTRPV3. These discrepancies in 2-APB binding between the human and mouse TRPV3 orthologs might be due to differential ligand affinities in different species. However, the role of site 2 and site 3 in 2-APB-dependent gating of mTRPV3 still remains to be electrophysiologically confirmed (*Singh et al., 2018*).

## The role of the distal CTD in activation of thermoTRPV

Following removal of 2-APB, the sensitized K169A, E751A, and W739A mutants retained residual activity after a prolonged washout of ligand (*Figure 5A*). This residual current, which might result from either an extremely slow 2-APB off-rate or constitutive activity at +60 mV, was sensitive to the pore blocker ruthenium red (RuR) (*Figure 5B*, *Figure 5—source data 1*). A voltage step from 0 to +60 mV prior to application of ligand elicited RuR sensitive currents (*Figure 5C,D*, *Figure 5—source data 1*) not seen in wild-type TRPV3 channels, suggesting that these mutants are inherently voltage gated in the absence of ligand. Notably, the G-V curve of the K169A and E751A mutants shows that voltage can directly activate the mutant channels (*Figure 5E*, *Figure 5—source data 1*), which is in stark contrast to the wild-type TRPV3 in which voltage is not sufficient for channel activation (*Hu et al., 2009*; *Xu et al., 2002*). Interestingly, the sensitized phenotype and voltage-dependent activation of the K169A and E751A TRPV3 bear resemblance to the behavior of the wild-type TRPV1 (*Gunthorpe et al., 2000*; *Voets et al., 2004*; *Sánchez-Moreno et al., 2018*), implying a mechanistic link in the activation of these two channels. A sequence comparison of thermoTRPV

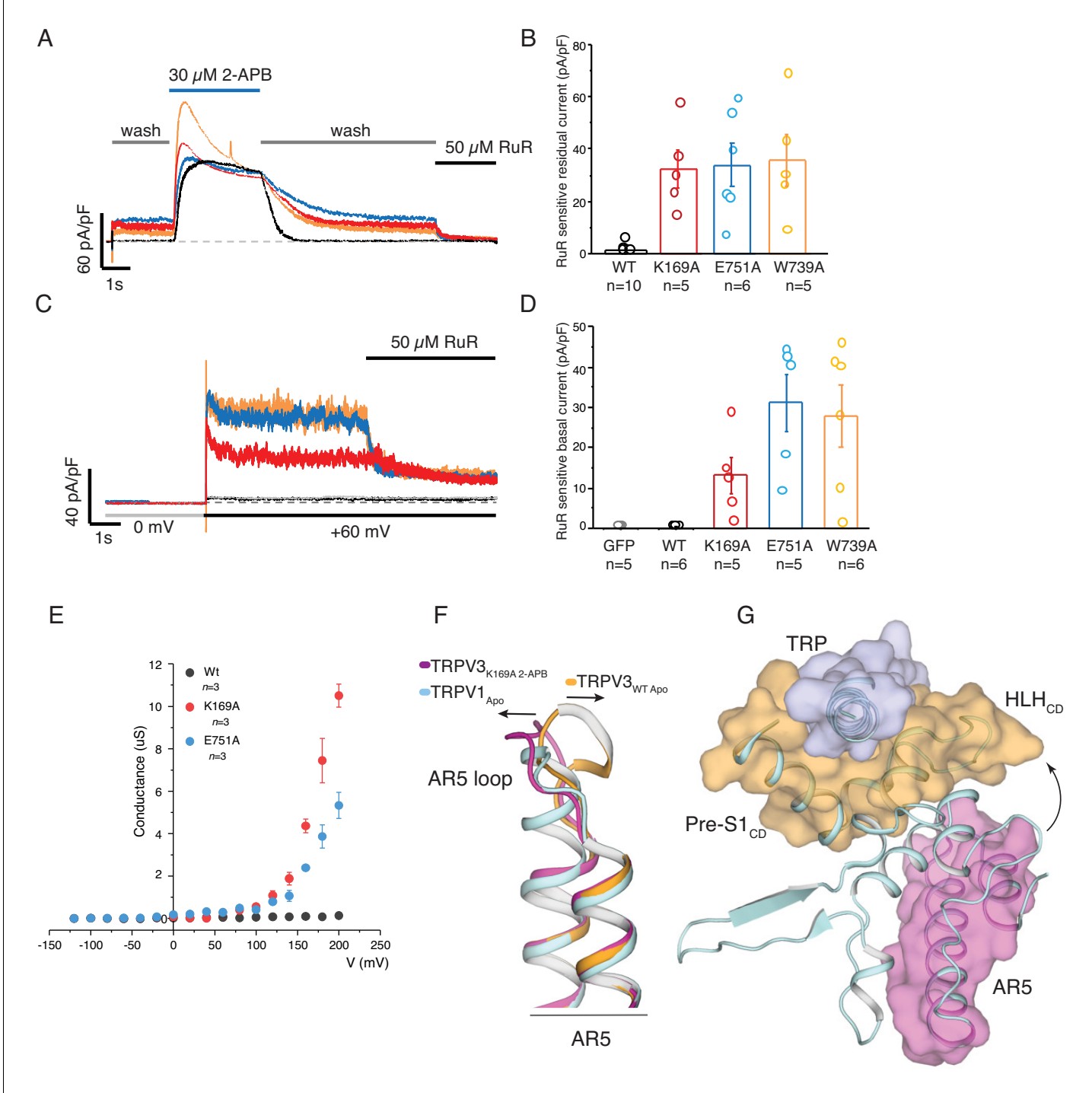

**Figure 5.** Parallels between TRPV3$_{K169A}$ and wild-type TRPV1. (**A**) Representative whole-cell recording at +60 mV of WT (*black*), K169A (*red*), E751A (*blue*), W739A (*orange*) immediately following 2-APB sensitization protocol, with perfusion protocol of 5 s wash, followed by 15 s 30 μM 2-APB, prolonged 30 s wash and the residual current was blocked by 5 s application of 50 μM ruthenium red (RuR). The scale bars units are in current density (pA/pF). (**B**) Graphical representation of RuR sensitive residual current at the end of the recording in (**A**) (WT: n = 10 biologically independent experiments; K169A: n = 5 biologically independent experiments; E751A: n = 6 biologically independent experiments, W739A: n = 5 biologically independent experiments). (**C**) Basal current activity before application of ligand was determined by the blocking the current following a voltage step from 0 to +60 mV with 50 μM RuR. (**D**) Graphical representation of the mean blocked current density from the protocol in (**C**) (GFP control: n = 5 biologically independent experiments; WT: n = 6 biologically independent experiments; K169A: n = 5 biologically independent experiments; E751A: n = 5 biologically independent experiments, W739A: n = 6 biologically independent experiments). (**E**) Conductance voltage relation of WT, K169A, and E751A in the absence of ligand determined from peak tail current elicited from a −160 mV post step pulse following a voltage step between −120

*Figure 5 continued on next page*

Figure 5 continued

to +200 mV (Δ 20 mV). The conductance of both mutants fail to saturate at +200 mV while WT channels are nonconductive at the tested voltages (WT: n = 3 biologically independent experiments; K169A: n = 3 biologically independent experiments; E751A: n = 3 biologically independent experiments). (F) Alignment of AR5 from TRPV3$_{WT}$ (orange), TRPV3$_{K169A\ 2APB}$ (magenta) and TRPV1 (light cyan). The AR5 loop of TRPV1 assumes a conformation similar to that of TRPV3$_{K169A\ 2APB}$. (G) The AR5 loop and the HLH$_{CD}$ are within interaction distance in TRPV1.

DOI: https://doi.org/10.7554/eLife.47746.023

The following source data and figure supplement are available for figure 5:

**Source data 1.** This spreadsheet contains the data used to calculate the ruthenium red sensitive current plots in *Figure 5B and 5D*; voltage step data used to calculate conductance plot in *Figure 5E*.
DOI: https://doi.org/10.7554/eLife.47746.025
**Figure supplement 1.** Sequence alignment of rat TRPV1 (rTRPV), rabbit TRPV2 (rabTRPV2), human TRPV3 (hTRPV3) and xenopus TRPV4 (xTRPV4).
DOI: https://doi.org/10.7554/eLife.47746.024

channels shows that the cytoplasmic inter-protomer interface is conserved and, remarkably, the distal CTD region in these channels exhibits high helical propensity (*Figure 5—figure supplement 1*). In order to examine the role of the distal CTD in the gating of thermoTRPVs, we inspected the cryo-EM map and model of the apo TRPV1 channel (*Liao et al., 2013*). Interestingly, we found striking similarities between the apo TRPV1 channel (PDB ID 3J5P) and the TRPV3$_{K169A-2APB}$ structure. Firstly, the conformation of the loop of AR5 in apo TRPV1 is similar to that observed in TRPV3$_{K169A-2APB}$ (*Figure 5F*). In addition, the cryo-EM density for the distal CTD, despite being poorly resolved, is compatible with a helical conformation of this region (*Figure 4—figure supplement 5*). Consequently, apo TRPV1 apparently exhibits tight coupling between the loop of AR5, the HLH$_{CD}$ and the TRP domain (*Figure 5G*). Furthermore, the pore lining S6 helix of apo TRPV1 adopts a π-helical turn (*Liao et al., 2013*). Therefore, it appears that the cytoplasmic and transmembrane domains in the TRPV1 channel are tightly connected even in the absence of stimuli, possibly explaining the absence of a sensitizing phenotype in this channel as well as its ability to be activated by voltage. Intriguingly, the K155A mutation in TRPV1, analogous to K169A in TRPV3, reduces ligand-induced desensitization of TRPV1 (*Lishko et al., 2007*; *Joseph et al., 2013*) and replacement of the distal CTD in TRPV1 with that of TRPV3 produces channels that do not desensitize upon repeated application of heat (*Joseph et al., 2013*). By contrast, deletion of the TRPV1 CTD helix produces non-functional channels, indicating that the distal CTD and the cytoplasmic inter-protomer interface is critical for gating and activation of TRPV channels (*Joseph et al., 2013*). The non-sensitized closed structures (*Zubcevic et al., 2018a*; *Zubcevic et al., 2018b*) of the sensitizing TRPV2 and TRPV3 (*Liu et al., 2011*; *Liu and Qin, 2016*) channels, possess CTDs in a coil conformation and a lower degree of coupling between the cytoplasmic and transmembrane domains (*Figure 4—figure supplement 5*). Therefore, our structural and functional analyses support this novel hypothesis that the distal CTD acts as a conformational switch in thermoTRPVs, which is imperative for coupling between the cytoplasmic and transmembrane domains and consequently also for channel activation.

## Discussion

The majority of the thermoTRPV channel structure resides in the cytoplasm and it has long been known that mutations in the cytoplasmic domains of these channels profoundly affect channel function. However, the role of these domains in thermoTRPV gating has been unclear since recent mechanistic studies of thermoTRPV channels have largely focused on the transmembrane domains. To the best of our knowledge, this study offers an unprecedented mechanistic insight into the involvement of the cytoplasmic domains in thermoTRPV gating as well as a potential explanation for the subtype specific ligand- and heat-dependent sensitization and activation phenotypes. The combination of our structural data and functional experiments provides evidence that hysteresis, sensitization, and activation of TRPV3 involves increased coupling between structural elements from the cytoplasmic (ARD and CD) and the transmembrane domains (TRP and S6) is triggered by a coil-to-helix transition in the distal CTD. The distal CTD appears to play a critical role akin to a binary switch in this coupling process. When the channel is in the naive closed state (C$_0$) the distal CTD adopts a coil ('off') conformation which is stretched around β$_{CD}$ via a salt-bridge 'hook' between K169 in the ARD and E751 and D752 in the CTD. When the salt-bridge is disrupted and the distal CTD 'unhooked', the coil

readily springs into a helical ('on') conformation, engages the loop of AR5, instigating a sequence of events that increase coupling in the sensitized $C_1$ state that is prerequisite for opening (*Figure 6*). We propose that this switch-like process involving a coil-to-helix secondary structure transition of the distal CTD is responsible, at least in part, for the use-dependent irreversible sensitization of TRPV3. In agreement with our previous report (*Zubcevic et al., 2018b*), the α-to-π helical transition in the pore lining S6 helix also forms a part of the use-dependent sensitization trajectory. We posit that the α-to-π helical transition in S6 constitutes an early and integral conformational transition leading to sensitization of the wild-type TRPV3 because it could be captured upon chemical sensitization of the channel, unlike the coil-to-helix transition (*Zubcevic et al., 2018b*). Because the coil-to-helix transition involves large rearrangements in secondary structure and changes in inter-protomer interactions, it is likely that it needs to surmount a larger energy barrier than the α-to-π transition in S6.

Our structural and functional analyses suggest that the unique cytoplasmic inter-protomer interface, which exists in all thermoTRPV channels, is a determinant of their distinct physiological and biophysical properties with the distal CTD acting as a switch between functional states in channel gating. The cytoplasmic inter-protomer interface and its coupling through to the transmembrane domains via the distal CTD, the ARD, CD, the TRP domain and S6 is conserved in thermoTRPV channels (*Figure 4—figure supplement 5*, *Figure 5—figure supplement 1*). The differences in the initial conformational state of the distal CTD from different thermoTRPV subtypes appear to correlate with their distinct sensitization and activation properties. The coil CTD, or 'off', conformation in the sensitizing TRPV2 and TRPV3 keeps channels in conformations associated with the closed $C_0$ state. By contrast, the CTD of the non-sensitizing TRPV1 is consistent with a helical 'on' conformation associated with the primed $C_1$ closed state. This proposes that the coupling network between the cytoplasmic and transmembrane domains can be modulated in a subtype-specific manner. The concept of channel gating via the cytoplasmic switch provides a framework for further examination of both

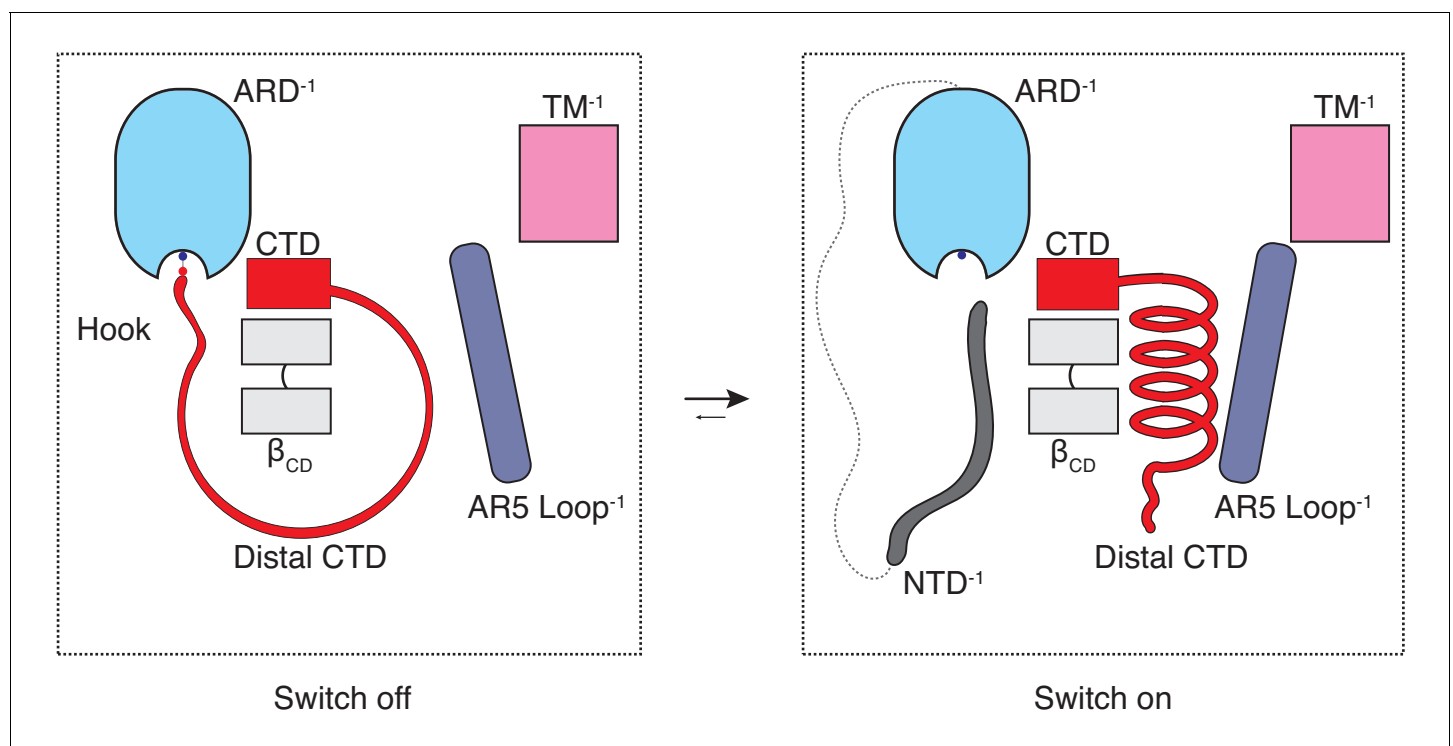

**Figure 6.** The CTD-mediated 'switch' gating mechanism. In the naive closed state ($C_0$), the distal CTD is stretched around the $\beta_{CD}$ via a salt bridge 'hook' interaction with the ARD and the switch is 'off'. When the CTD is 'unhooked' from the ARD, it undergoes a coil-to-helix transition (the switch turns 'on') which leads to a conformational change in the loop of AR5 and consequently to increased coupling between the cytoplasmic and transmembrane domains, which is prerequisite for channel opening ($C_1$).
DOI: https://doi.org/10.7554/eLife.47746.026

ligand and heat dependent gating of thermoTRPV channels, which could be extended to other ion channel families.

# Materials and methods

## Key resources table

| Reagent type (species) or resource | Designation | Source or reference | Identifiers | Additional information |
|---|---|---|---|---|
| Cell line (*E. coli*) | DH10Bac | ThermoFisher Scientific | 10361012 | |
| Cell line (*Spodoptera frugiperda*) | Sf9 | ATCC | CRL-1711 | RRID:CVCL_0549 |
| Cell line (*Homo sapiens*) | HEK293T | ATCC | CRL-11268; Lot Number 62312975 | RRID:CVCL_0063 |
| Cell media component | Dulbecco's Modified Eagle's Medium (DMEM) - low glucose | Gibco | 11885–084 | |
| Cell media component | Heat Inactivated Fetal Bovine Serum | Gibco | 10082–139 | |
| Cell media component | Anti-Anti (Antibiotic-Antiycotic) | Gibco | 15240–062 | |
| Recombinant DNA reagent | human TRPV3 | Genscript | Pubmed Gene ID: 162514 | |
| Recombinant DNA reagent | Bac-to-Bac Baculovirus Expression System | ThermoFisher Scientific | 10359016 | |
| Recombinant DNA reagent | FuGene6 | Promega | E2691 | |
| Chemical compound, drug | $n$-dodecyl-$\beta$-d-maltopyranoside(DDM) | Anatrace | D310 | |
| Chemical compound, drug | Cholesteryl Hemisuccinate | Anatrace | CH210 | |
| Chemical compound, drug | PMAL-C8 | Anatrace | P5008 | |
| Chemical compound, drug | TRIS | Fisher Scientific | BP152 | |
| Chemical compound, drug | NaCl | Fisher Scientific | S271 | |
| Chemical compound, drug | $CaCl_2$ | Fisher Scientific | C70 | |
| Chemical compound, drug | KCl | Sigma Aldrich | P9333 | |
| Chemical compound, drug | $MgCl_2$ | Sigma Aldrich | M8266 | |
| Chemical compound, drug | 4-(2-hydroxyethyl)—1-piperazineethanesulfonic acid (HEPES) | Sigma Aldrich | H3375 | |
| Chemical compound, drug | NaOH | Sigma Aldrich | S5881 | |
| Chemical compound, drug | CsCl | Sigma Aldrich | C3139 | |
| Chemical compound, drug | Ethylene glycol-bis (2-aminoethylether)-N,N,N',N'-tetraacetic acid (EGTA) | Sigma Aldrich | E4378 | |

*Continued on next page*

*Continued*

| Reagent type (species) or resource | Designation | Source or reference | Identifiers | Additional information |
|---|---|---|---|---|
| Chemical compound, drug | CsOH solution | Sigma Aldrich | 232041 | |
| Chemical compound, drug | 2-Aminoethyl diphenylborinate (2-APB) | Sigma Aldrich | D9754 | |
| Chemical compound, drug | D-Camphor | Sigma Aldrich | W223018 | |
| Chemical compound, drug | Dimethyl sulfoxide (DMSO) | Sigma Aldrich | D2650 | |
| Chemical compound, drug | leupeptin | GoldBio | L-010 | |
| Chemical compound, drug | pepstatin | GoldBio | P-020 | |
| Chemical compound, drug | aprotinin | GoldBio | A-655 | |
| Chemical compound, drug | DNase I | GoldBio | D-301 | |
| Chemical compound, drug | β-mercapto ethanol | Sigma Aldrich | M3148 | |
| Chemical compound, drug | PMSF | Sigma Aldrich | P7626 | |
| Chemical compound, drug | anti-FLAG resin | Sigma Aldrich | A4596 | |
| Chemical compound, drug | Bio-Beads SM-2 | BioRad | 152–8920 | |
| Chemical compound, drug | 1-palmitoyl-2-oleoyl -*sn*-glycero-3-phosphocholine (POPC) | Avanti Polar Lipids | 850457C | |
| Chemical compound, drug | 1-palmitoyl-2-oleoyl -*sn*-glycero-3-phosphoethanolamine (POPE) | Avanti Polar Lipids | 850757C | |
| Chemical compound, drug | 1-palmitoyl-2-oleoyl -*sn*-glycero-3-phospho-(1'-*rac*-glycerol) (POPG) | Avanti Polar Lipids | 840457C | |
| Software, algorithm | MotionCor2 | *Zheng et al., 2017* | http://msg.ucsf.edu/ em/software/motioncor2. html | RRID:SCR_016499 |
| Software, algorithm | GCTF | *Zhang, 2016* | https://www.mrc-lmb .cam.ac.uk/kzhang/ | RRID:SCR_016500 |
| Software, algorithm | RELION 3.0 | *Zivanov et al., 2018* | https://www2.mrc-lmb .cam.ac.uk/relion/ | RRID:SCR_016274 |
| Software, algorithm | Coot | *Emsley and Cowtan, 2004* | https://www2.mrc-lmb. cam.ac.uk/personal/ pemsley/coot/ | RRID:SCR_014222 |
| Software, algorithm | Phenix | *Adams et al., 2010* | http://phenix-online.org/ | RRID:SCR_014224 |
| Software, algorithm | Molprobity | *Chen et al., 2010* | http://molprobity.biochem .duke.edu/index.php | RRID:SCR_014226 |
| Software, algorithm | UCSF Chimera | *Pettersen et al., 2004* | https://www.cgl.ucsf. edu/chimera/ | RRID:SCR_004097 |
| Software, algorithm | Pymol | Shrödinger LLC | https://pymol.org/2/ | RRID:SCR_000305 |

*Continued on next page*

*Continued*

| Reagent type (species) or resource | Designation | Source or reference | Identifiers | Additional information |
|---|---|---|---|---|
| Software, algorithm | pClamp10 | Molecular Devices | RRID:SCR_011323 | |
| Software, algorithm | OriginPro 2016 | OriginLab Corp. | RRID:SCR_014212 | |
| Software, algorithm | Microsoft Excel 2010 | Microsoft | RRID:SCR_016137 | |
| other | Whatman No. one filter paper | Sigma Aldrich | WHA1001325 | |
| Other | UltrAuFoil R1.2/1.3 300-mesh grid | Electron Microscopy Sciences | Q350AR13A | |
| Other | Cryo-electron microscopy structure of the human TRPV3 channel | *Zubcevic et al., 2018a* | PDB ID 6MHO | *Zubcevic et al., 2018b* |
| Other | Cryo-electron microscopy structure of the human TRPV3 channel | *Zubcevic et al., 2018a* | EMDB ID EMD-9115 | *Zubcevic et al., 2018b* |

## Expression and purification of human TRPV3

A full-length human TRPV3 construct, containing mutations T96A (*Zubcevic et al., 2018b*) and K169A was cloned into a pFastBac vector in frame with a FLAG affinity tag, and baculovirus was produced according to manufacturers' protocol (Invitrogen, Bac-to-Bac). Sf9 insect cells (ATCC), infected with baculovirus at a density of $1.3 \times 10^6$ cells ml$^{-1}$, were grown for 72 hr at 27° C in an orbital shaker. Cell pellets were collected and resuspended in buffer A (50 mM TRIS pH 8, 150 mM NaCl, 1 µg ml$^{-1}$ leupeptin, 1.5 µg ml$^{-1}$ pepstatin, 0.84 µg ml$^{-1}$ aprotinin, 0.3 mM PMSF, 14.3 mM β-mercaptoethanol, and DNAseI) before lysis by sonication (3 × 30 pulses). Lysed cells were solubilized in 40 mM dodecyl β-maltoside (DDM, Anatrace) and 4 mM Cholesteryl Hemisuccinate Tris salt (CHS, Anatrace) at 4° C for 1 hr. Insoluble material was removed by centrifugation (8,000 *g*, 30 min), and anti-FLAG resin was added to the supernatant for 1 hr at 4° C. Following binding, the anti-FLAG resin was transferred to a Biorad column at 4° C and washed with 10 column volumes buffer B (50 mM TRIS pH8, 150 mM NaCl, 1 mM DDM, 0.1 mM CHS, 10 mM DTT) and the protein eluted in buffer C (50 mM TRIS pH 8, 150 mM NaCl, 1 mM DDM, 0.1 mM CHS, 0.1 mg ml$^{-1}$ 3:1:1 1-palmitoyl-2-oleoyl-*sn*-glycero-3-phosphocholine (POPC), 1-palmitoyl-2-oleoyl-*sn*-glycero-3-phosphoethanolamine (POPE), 1-palmitoyl-2-oleoyl-*sn*-glycero-3-phospho-(1'-*rac*-glycerol) (POPG), 10 mM DTT, 10 mg ml$^{-1}$ FLAG peptide). Size exclusion chromatography was performed and the protein peak collected and mixed with Poly (Maleic Anhydride-alt-1-Decene) substituted with 3-(Dimethylamino) Propylamine (PMAL-C8, Anatrace) (1:10 w/w ratio) and incubated overnight at 4° C with gentle agitation. Detergent was removed with Bio-Beads SM-2 (15 mg ml$^{-1}$) for 1 hr at 4° C. The reconstituted protein was purified on a Superose 6 column at 4° C in buffer D (50 mM Tris pH8, 150 mM NaCl). Following size exclusion, the protein peak was collected and concentrated to 2–2.5 mg ml$^{-1}$. For the TRPV3$_{K169A\ 2-APB}$ sample, the protein was incubated with 1 mM 2-APB for ~3.5 min before blotting.

## Cryo-EM sample preparation

Cryo-EM grid preparation was performed similarly for each TRPV3 K169A specimen. 3 µl sample was dispensed on a freshly glow discharged (30 s) UltrAuFoil R1.2/1.3 300-mesh grid (Electron Microscopy Services), blotted for 3 s with Whatman No. one filter paper using the Leica EM GP2 Automatic Plunge Freezer at 23° C and >85% humidity and plunge-frozen in liquid ethane cooled by liquid nitrogen.

## Cryo-EM data collection

Data for TRPV3$_{K169A}$ and TRPV3$_{K169A\ 2-APB}$ was collected using the Titan Krios transmission electron microscope (TEM) operating at 300 keV using Gatan K2 Direct Electron Detector and a Falcon III

Direct Electron Detector operating in counting mode, respectively. The nominal magnification used for the TRPV2$_{K169A}$ sample was 130,000x corresponding to a physical pixel size of 1.06 Å/pixel. For the TRPV2$_{K169A\ 2-APB}$, the nominal magnification was 75,000x corresponding to a physical pixel size of 1.08 Å/pixel. For the TRPV3$_{K169A}$, 2385 movies (40 frames/movie) were collected using a 10 s exposure with an exposure rate of ~4.5 e⁻/pixel/s, resulting in a total exposure of 40 e⁻/Å (*Caterina et al., 1997*) and a nominal defocus range from −1.0 μm to −2.5 μm. For TRPV3$_{K169A\ 2-APB}$, 1984 movies were collected (30 frames/movie) with 60 s exposure and exposure rate of ~0.8 e⁻/pixel/s. The total exposure was of 42 e⁻/Å (*Caterina et al., 1997*) and a nominal defocus range from −1.25 μm to −3.0 μm.

## Reconstruction and refinement

*TRPV3$_{K169A}$* MotionCor2 (*Zheng et al., 2017*) was used to perform motion correction and dose-weighting on 2385 movies. Summed unweighted images were used for CTF determination using GCTF (*Zhang, 2016*). After motion correction and CTF determination, the dataset was pruned by removing micrographs which contained Figure of Merit (FoM) values of <0.05 and Astigmatism values > 1700. A set of 1596 particles was picked manually and subjected to reference-free 2D classification (k = 12, T = 2) which subsequently served as a template for automatic particle picking from the entire dataset. A stack of 452,388 particles were picked (binned 4 × 4 (4.24 Å/pixel, 64 pixel box size)) and subjected to reference-free 2-D classification (k = 45, T = 2) in RELION 3.0 (*Zivanov et al., 2018*). Classes displaying the most well-defined secondary structure features were selected (441,547 particles) and used in 3D refinement with no symmetry imposed (C1) and with the previously determined map for apo human TRPV3 (EMD-9115) filtered to 30 Å as a reference model. This resulted in an 8.7 Å 3D reconstruction, which was then used for re-extraction and re-centering of 1 × 1 binned particles (1.06 Å/pixel, 256 pixel box size). 3D classification (k = 6, T = 8) without imposed symmetry (C1) was performed on these particles, using a soft mask calculated from the full molecule. Class 4 (95,184 particles) possessed the most well-defined secondary structure elements and was chosen for further analysis. 3D auto-refinement of class four without symmetry imposed (C1) yielded a 4.6 Å 3D reconstructions. The particles were then subjected to Bayesian polishing as implemented in RELION 3.0. The shiny particles were input into 3D auto-refinement with a soft mask no imposed symmetry (C1), resulting in a 4.3 Å reconstruction. These particles were subjected to CTF refinement, followed by another round of 3D auto-refinement in C1 symmetry resulting in a 4.37 Å map. Visual inspection of the volume revealed the presence of four-fold symmetry, and therefore 3D auto-refinement was repeated with C4 symmetry imposed, resulting in the final 4.1 Å map.

*TRPV3$_{K169A\ 2-APB}$* 1984 movies were subjected to motion correction and dose-weighing using MotionCor2. The unweighted and summed images were used for CTF determination using GCTF. Micrographs with a Figure of Merit (FoM) values of <0.15 and Astigmatism values of >200 were removed. A set of 2099 particles was picked manually and subjected to reference-free 2D classification (k = 12, T = 2) which were used as a template for automatic particle picking from the entire dataset. This resulted in a stack of 1,174,521 particles (binned 4 × 4 (4.32 Å/pixel, 64 pixel box size)) and subjected to reference-free 2-D classification (k = 75, T = 2) in RELION 3.0 (*Zivanov et al., 2018*). Classes with the most well-defined structural features were picked (636,742 particles) and extracted (binned 1 × 1 (1.08 Å/pixel, 256 pixel box size)) before being subjected to 3D auto-refinement using the map of human apo TRPV3 (EMD-9115) filtered to 30 Å as a reference and with no symmetry imposed (C1), resulting in a 5.3 Å reconstruction. This was then subjected to 3D classification (k = 6, T = 8) which included particle alignment and with no symmetry imposed. Class 4 (136,814 particles) was selected and subjected to 3D auto-refinement (C1), yielding a 3.9 Å reconstruction. Further 3D classification (k = 2, T = 8) using a soft mask and without alignment or imposed symmetry separated a fraction of bad particles (57,808 particles) leaving a stack of 79,006 particles which were subjected to another round of 3D auto-refinement (C1). The resulting volume revealed four-fold symmetry and the 3D auto-refinement was therefore repeated with C4 symmetry imposed, yielding a 3.6 Å reconstruction. These particles were then subjected to Bayesian polishing and CTF refinement, resulting in a final reconstruction resolved to 3.59 Å. All resolution estimates were based on the gold-standard FSC 0.143 criterion (*Scheres and Chen, 2012*; *Chen et al., 2013*).

## Model building

The TRPV3$_{K169A}$ and TRPV3$_{K169A\ 2-APB}$ models were built directly into the cryo-EM electron density using the previously determined structure of the human TRPV3 in the apo form (PDB 6MHO) as a template. The models were first refined in real space in Coot (*Emsley and Cowtan, 2004*) and subsequently subjected to automated real space refinement using phenix.real_space_refine as implemented in the Phenix suite (*Adams et al., 2010*). The refinement was performed using global minimization and rigid body, with tight ideal geometry and secondary structure restraints. The refinement process was guided by the Molprobity server (http://molprobity.biochem.duke.edu/) (*Chen et al., 2010*). Analysis and structure illustrations were performed using Pymol (The PyMOL Molecular Graphics System, Version 2.0) and UCSF Chimera (*Pettersen et al., 2004*).

## Model comparisons and analysis

All structural alignments and measurements were performed using Pymol and UCSF Chimera. The correlation of the fit of the coil vs. the helix distal CTD into the mTRPV3 cryo-EM map (EMD-8921) was calculated as follows: the EMD-8921 map was segmented using the *Segment map* tool, as implemented in UCSF Chimera, and the density corresponding to the distal CTD was isolated and saved as a map file (Map$_{CTD}$). The coil CTD (PDB 6DVZ) and the helical CTD (TRPV3$_{K169A\ 2-APB}$) were isolated and saved as individual pdb files (CTD$_{coil}$ and CTD$_{helix}$) and placed in the Map$_{CTD}$. Using the *molmap* function in UCSF Chimera, simulated maps were generated for CTD$_{coil}$ and CTD$_{helix}$ at the same resolution as Map$_{CTD}$. Correlation was calculated between the CTD$_{coil}$/Map$_{CTD}$, and CTD$_{helix}$/-Map$_{CTD}$ using the *measure correlation* function in UCSF Chimera.

## Cell lines

HEK293T cells were purchased from ATCC with authentication records. Additional authentication was not performed prior to this study. Cells tested negative for mycoplasma contamination.

## Electrophysiology

HEK293T cells (62312975 – ATCC) were grown in DMEM supplemented with 10% FBS (Gibco), 1% penicillin/streptomycin (Gibco) and were sustained at 37°C in 5% $CO_2$. Cells between passage 10–30 grown in 40 mm wells were transiently transfected at ~50% confluency with plasmids encoding for either WT, K169A, E751A, W739A, W742A, K743A, R487A, R487W, E501G, Y540W, Y565A, H426A TRPV3 and green fluorescent protein (GFP) using FuGene6 (Promega). ~24 hr after transfection, cells were reseeded onto 12 mm round glass coverslips (Fisher) in 20 mm wells and used after 12–24 hr for electrophysiological measurements.

Voltage-clamp recording were performed in the whole-cell patch-clamp configuration with electrodes pulled from borosilicate glass capillaries (Sutter Instruments) with a final resistance of 2–5 MΩ. Electrodes were filled with an intracellular solution containing (in mM) 150 CsCl, 1 $MgCl_2$, 10 HEPES, 5 EGTA, and adjusted to pH 7.2 (CsOH). Glass coverslips with adherent transfected cells were placed into an open bath chamber (RC-26G, Warner Instruments) with an extracellular wash solution containing (in mM) 140 NaCl, 5 KCl, 1 $MgCl_2$, 10 HEPES at pH 7.4 (NaOH). Extracellular wash solutions were used to make solutions containing; 2-aminoethoxydiphenyl borate (2-APB) (Sigma) (prepared daily from DMSO stocks (1 M) stored at −80° C; final DMSO 0.03%), ruthenium red (RuR) (Sigma) (prepared daily from water stock (10 mM) stored at −80° C; final DMSO 0.03%), and D-camphor (Sigma) (prepared daily as previously described [*Xu et al., 2005*] from DMSO stocks (2 M) stored at −80° C). Solutions were focally applied to patched calls with a pressurized perfusion system (BPS-8, ALA Scientific Instruments). Current responses were low-pass filtered at 2 kHz (Axopatch 200B), digitally sampled at 5–10 kHz (Digidata 1440A), converted to digital files in Clampex10.7 (Molecular Devices) and stored on an external hard drive for offline analyses (Clampfit10.7, Molecular Devices; Excel 2010, Microsoft Office; OriginPro 2016, OrginLab Corp).

2-APB sensitization experiments were performed as previously described (*Zubcevic et al., 2018b*). Briefly, a 30 s continuously repeating protocol (holding potential of +60 mV) was used in which cells were first perfused with extracellular wash for 1 s, followed by a 15 s application of 30 µM 2-APB extracellular solution, preceded by 14 s of wash solution. Recordings that displayed little to no current response after ~25–30 rounds of 2-APB stimulation were stopped.

The remaining residual sensitized current following washout of 30 µM 2-APB was measured immediately following the 2-APB sensitization protocol. Sensitized cells underwent a protocol (holding potential of +60 mV) in which extracellular wash was perfused for 10 s, followed by 15 s of 30 µM 2-APB, then washed again for 30 s, proceeded by application of 50 µM RuR for 10 s. The RuR-sensitive residual current was calculated by the difference between the second wash ($I_{wash2}$) and RuR ($I_{RuR}$) current amplitude ($I_{wash2} - I_{RuR}$) and is reported as current density.

Spontaneous basal channel activity was assessed prior to 2-APB-induced sensitization. After achieving the whole-cell configuration, cells were held at 0 mV for 5 s and underwent a voltage step +60 mV for 10 s. 5 s after the voltage step to +60 mV, the cell was perfused with 50 µM RuR for the last 5 s of the protocol. RuR sensitive basal currents were calculated as the difference between the measured wash ($I_{wash}$) and RuR ($I_{RuR}$) current ($I_{wash} - I_{RuR}$) amplitudes at +60 mV and are reported as current density.

Whole-cell voltage step protocol from −120 to +200 mV (Δ 20 mV, 500 ms) was immediately followed by a −160 mV post-test pulse for 200 ms with a holding potential of −60 mV. The peak tail current amplitudes from the −160 mV post-test pulse were used to calculate the corresponding conductance amplitude for each voltage step. Since the tail currents from the +200 mV step did not result in saturating tail current amplitudes, G/Gmax curves were not constructed and the data was plotted as the conductance amplitudes versus the step voltage.

Hysteresis and sensitization were assessed by measuring changes in the $EC_{50}$ of 2-APB sensitivity following consecutive dose-response rounds. Cells underwent a recording protocol (holding potential of +60 mV) that started with a 1 s wash, followed by 15 s application of a single concentration of 2-APB (3, 10, 30, 50, 100 or 300 µM; final DMSO 0.03%), followed by 15 s of wash and was repeated for each concertation, in order from lowest to highest concentration for each dose-response round. Cells were subjected to three consecutive rounds of this dose-response protocol.

2-APB binding site testing experiments were performed with a continuously repeating voltage ramp protocol (holding potential 0 mV, 400 ms voltage ramp from −60 to +60 mV) elicited every 5 s. Cells were first perfused for 30 s with 30 µM 2-APB, followed by 300 µM 2-APB, and finally 10 mM camphor, with a 30 s wash between each application of ligand. The ratio between both 30 and 300 µM 2-APB to the camphor current response at +60 mV was calculated. Leak was assessed at the end of the recording with application of 50 µM RuR.

Sensitization was characterized by the ratio of the response to 2-APB during the first ($I_0$) and maximum current ($I_{max}$) response ($I_{max}/I_0$) calculated as the mean from each biologically independent experiment.

*Data Analysis* 2-APB sensitization parameters were evaluated by the stimulation dependent increase in the peak current amplitude measured at the end of each 2-APB exposure as previously described (*Zubcevic et al., 2018b*). Briefly, peak current amplitudes ($I$) from each individual stimulation was normalized to the maximum peak current ($I_{max}$) amplitude, and the fractional current ($I/I_{max}$) of each stimulation was plotted by stimulation number for each individual recording.

The relative extent of sensitization was characterized by the increase in current amplitude obtained during the first ($I_0$) and maximum current ($I_{max}$) stimulation ($I_{max}/I_0$) and calculated as the mean from each biologically independent experiment. The first five 2-APB stimulations per recordings was also plotted as current density for comparison between conditions.

2-APB dose-response data for each individual round were fit with the Hill equation from biologically independent experiments. The average $EC_{50}$ values from each fit was calculated for each dose-response. The averaged normalized current response from each 2-APB concertation (3, 10, 30, 50, 100, 300 µM) per round were averaged and fit with the Hill equation to calculate the corresponding $EC_{50}$ and Hill coefficient ($n_H$) for each construct tested.

## Acknowledgements

Cryo-EM data were collected at the Shared Materials Instrumentation Facility at Duke University as part of the Molecular Microscopy Consortium, and at the National Cryo-EM Facility (NCEF). Sample screening was performed at National Institute for Environmental Health Sciences (NIEHS). We thank Alberto Bartesaghi at Duke for a pre-processing interface, and Ulrich Baxa and Thomas Edwards for assistance with data collection at NCEF. We also thank Jorg Grandl and Huanghe Yang for useful discussions during manuscript preparation. Funding: This work was supported by the National

Institutes of Health (R35NS097241 to S-YL) and by the National Institutes of Health Intramural Research Program; US National Institute of Environmental Health Science (ZIC ES103326 to MJB). This research was, in part, supported by the National Cancer Institute's National Cryo-EM Facility at the Frederick National Laboratory for Cancer Research under contract HSSN261200800001E.

## Additional information

### Funding

| Funder | Grant reference number | Author |
| --- | --- | --- |
| National Institutes of Health | Intramural Research Program | Mario J Borgnia |
| National Institute of Environmental Health Sciences | ZIC ES103326 | Mario J Borgnia |
| National Institute of Neurological Disorders and Stroke | R35NS097241 | Seok-Yong Lee |

The funders had no role in study design, data collection and interpretation, or the decision to submit the work for publication.

### Author contributions

Lejla Zubcevic, William F Borschel, Conceptualization, Data curation, Formal analysis, Writing—original draft, Writing—review and editing; Allen L Hsu, Data curation, Validation, Writing—review and editing; Mario J Borgnia, Supervision, Funding acquisition, Validation, Writing—review and editing; Seok-Yong Lee, Conceptualization, Resources, Supervision, Funding acquisition, Validation, Writing—original draft, Project administration, Writing—review and editing

### Author ORCIDs

Lejla Zubcevic https://orcid.org/0000-0002-1884-9235
William F Borschel https://orcid.org/0000-0003-4064-9026
Allen L Hsu https://orcid.org/0000-0003-2065-3802
Mario J Borgnia https://orcid.org/0000-0001-9159-1413
Seok-Yong Lee https://orcid.org/0000-0002-0662-9921

### Decision letter and Author response

Decision letter https://doi.org/10.7554/eLife.47746.037
Author response https://doi.org/10.7554/eLife.47746.038

## Additional files

### Supplementary files

• Transparent reporting form
DOI: https://doi.org/10.7554/eLife.47746.027

### Data availability

Cryo-EM data and structural models are deposited in the EMDB and RCSB, respectively with the following codes: EMD-20192, PDB: 6OT2 and EMD-20194, PDB: 6OT5

The following datasets were generated:

| Author(s) | Year | Dataset title | Dataset URL | Database and Identifier |
| --- | --- | --- | --- | --- |
| Zubcevic L, William F Borschel, Allen L Hsu, Mario J Borgnia, Lee S-Y | 2019 | Structure of the TRPV3 K169A sensitized mutant in the presence of 2-APB at 3.6 A resolution | http://www.ebi.ac.uk/pdbe/entry/emdb/EMD-20194 | Electron Microscopy Data Bank, EMD-20194 |
| Zubcevic L, William | 2019 | Structure of the TRPV3 K169A | http://www.rcsb.org/ | Protien Data Bank, |

| | | | | |
|---|---|---|---|---|
| F Borschel, Allen L Hsu, Mario J Borgnia, Lee S-Y | | sensitized mutant in the presence of 2-APB at 3.6 A resolution | structure/6OT5 | 6OT5 |
| Zubcevic L, William F Borschel, Allen L Hsu, Mario J Borgnia, Lee S-Y | 2019 | Structure of the TRPV3 K169A sensitized mutant in apo form at 4.1 A resolution | http://www.ebi.ac.uk/pdbe/entry/emdb/EMD-20192 | Electron Microscopy Data Bank, EMD-20192 |
| Zubcevic L, William F Borschel, Allen L Hsu, Mario J Borgnia, Lee S-Y | 2019 | Structure of the TRPV3 K169A sensitized mutant in apo form at 4.1 A resolution | http://www.rcsb.org/structure/6OT2 | Protein Data Bank, 6OT2 |

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
