## [Decision Letter]

Thank you for submitting your article "Regulatory switch at the cytoplasmic interface controls TRPV channel gating" for consideration by *eLife*. Your article has been reviewed by Richard Aldrich as the Senior Editor, Kenton Swartz as the Reviewing Editor, and two reviewers. The following individuals involved in review of your submission have agreed to reveal their identity: Youxing Jiang (Reviewer #1) and Andres Jara-Oseguera (Reviewer #2).

The reviewers have discussed the reviews with one another and the Reviewing Editor has drafted this decision to help you prepare a revised submission.

Summary:

Activation of the TRPV3 channel involves an effectively irreversible step possibly involving a large energy barrier, which results in prominent hysteresis in activation and sensitized responses to the agonist 2-APB and heat once the barrier has been overcome. Whereas hysteresis in TRPV3 is very strong, it can also be observed to a lesser degree in TRPV2, whereas agonist-induced sensitization in TRPV1 is much less pronounced and does not influence responses to heat in this channel. The molecular mechanism responsible for this in unknown. In the present manuscript, Zubcevic and collaborators provide strong evidence drawing from structural and functional data to support a mechanism that explains sensitization in TRPV3, and the possible origin of the differences in sensitization between TRPV1, TRPV2 and TRPV3 channels.

The authors identified a series of inter- and intra-protomer contacts involving the membrane-proximal N-terminal domain and the C-terminal domain following the TRP domain in their closed human TRPV3 structure. When a key salt-bridge between the distal C-terminal domain (CTD) and the N-terminal ankyrin of the neighboring subunit is disrupted, channels exhibit a sensitized phenotype in functional experiments, and the structure of the mutant (K169A) exhibits pronounced secondary and tertiary structural re-arrangements in this region, involving a loop-to-helix switch in the CTD domain. An alanine substitution to one additional residue, W739A, resulted in a sensitized phenotype similar to K169A and E751A. In addition, disrupting the interactions specific to the TRPV3K169A (sensitized) structure resulted in channels with decreased sensitization.

A structure combining the presence of the agonist 2-APB and the K169A mutation results in similar conformation changes in the cytosolic domains as K169A, with additional conformational changes in the transmembrane region, including the pore domain, resulting in conformation that is consistent with ion-conduction. The authors provide further evidence using site-directed mutagenesis to support the hypothesis that 2-APB interacts at precisely the region where the sensitizing switch is located. The structural differences between the three structures provide a compelling mechanism that explains how the changes in the cytosolic domain could lead to channel opening upon 2-APB binding. Finally, the authors compare the density maps of TRPV2 and TRPV1 in the apo state with the conformational changes at the cytosolic switch observed in TRPV3, and show that whereas apo TRPV2 is consistent with the WT TRPV3 structure, that of TRPV1 is more similar to the K169A+2-APB TRPV3 structure. This is an elegant study that provides strong functional and structural support for an important mechanism for TRPV channel gating. We have only a few minor points that could further improve the manuscript.

1) "Analogous sensitization upon stimulation with capsaicin or heat has not been observed in TRPV1" (Introduction) – this statement is not entirely accurate; 2-APB-dependent sensitization has been observed in the case of TRPV1, albeit to a much lesser extent than for TRPV3, and more importantly, without affecting subsequent responses to heat, whereas capsaicin exhibits no sensitization (Liu and Qin, 2016). However, a more recent study has provided strong evidence that heat activation of TRPV1 is associated with pronounced hysteresis (Sanchez-Moreno et al., 2018).

2) We suggest using the term "sensitized" instead of "pre-sensitized" for K169A and related mutants, as their phenotype coincides with that of sensitized channels. We find the term pre-sensitized can be confounding.

3) "due to impaired hysteresis" (subsection “Conformational changes of the distal CTD lead to rearrangements in the cytoplasmic inter subunit interface”) – may be a bit awkward. We suggest using "impaired sensitization" instead, as hysteresis is the phenomenological consequence of sensitization.

4) Can the pi-helix in the S6 be unequivocally assigned at the resolution for each structure?

5) In the dose-response curves for the mutants with decreased activity, the apparent Kd values obtained are probably an overestimation, since the curves don't saturate. The authors should mention this, which in fact strengthens their conclusions.

6) The authors should use the experimental maps to provide a clear illustration of the conformational changes occurring at the switch such as those in Figures 2B,C, Figure 3C and Figure 4D, in which surface representations were used.

7) The mouse TRPV3 channel mutation of residue N412 has been shown to have a large effect on sensitization (Liu and Qin, 2017). Is there any structural difference coming from the structures that could also explain the effects of this mutant?

---

## [Author Response]

[…] We have only a few points that could further improve the manuscript.1) "Analogous sensitization upon stimulation with capsaicin or heat has not been observed in TRPV1" (Introduction) – this statement is not entirely accurate; 2-APB-dependent sensitization has been observed in the case of TRPV1, albeit to a much lesser extent than for TRPV3, and more importantly, without affecting subsequent responses to heat, whereas capsaicin exhibits no sensitization (Liu and Qin, 2016). However, a more recent study has provided strong evidence that heat activation of TRPV1 is associated with pronounced hysteresis (Sanchez-Moreno et al., 2018).

As the reviewers point out, “sensitization” of TRPV1 is only observed in response to consecutive application of 2-APB. However, this use-dependent increase in activity is stimulus specific, as 2-APB sensitization does not affect the response to heat or capsaicin (Liu and Qin, 2016). Furthermore, this study did not examine if this change in 2-APB activation is the result of irreversible changes (hysteresis) in gating. Therefore, we chose not to include this in the manuscript as the mechanism of 2-APB use-dependence in TRPV1 is not clear and hysteretic behavior was not confirmed.

Sanchez-Moreno and colleagues did indeed show that heat activation of TRPV1 is associated with hysteresis. However, hysteresis of heat dependent gating leads to a progressive decrease in response, resulting in irreversible entry into the inactivated state. These findings are consistent with our study; prolonged 2-APB stimulation of both the WT TRPV3 and the sensitized mutants results in a steady decrease of the current amplitude, indicative of inactivation. While inactivation of WT channels occurs only after prolonged repeated application of 2-APB (~35 – 45 cycles; Figure 1C-D), inactivation develops more quickly in the sensitized mutants (~10 cycles; Figure 1E-H). As noted in the discussion, we believe that the shape of the current vs stimulation number plot for TRPV channels correlates with the starting conformation of the CTD switch: the thermoTRPV channels which possess a starting “coil” CTD conformation are initially resting in the C_0_ state. Following consecutive stimulations, the current becomes larger as the C_0_ state occupancy decreases while the C_1_ and O state occupancy increases. As the O state occupancy increases, channels begin to inactivate, leading to a decay of the current response. However, channels possessing a starting “helix” CTD conformation (i.e. the K169A mutant and TRPV1), will initially be in the C_1_ state at rest. The initial application of a stimulus results in faster activation (C_1_ → O) as the channels bypass the requisite C_0_ → C_1_ transition, resulting in entry to the inactivated state from the O state. Our TRPV3 data and the TRPV1 study done by Sanchez-Moreno et al. hint at the existence of multiple stimuli-dependent inactivated states in these closely related channels, but he nature of inactivation in TRPV3 merits a separate in-depth study that can shed further light on the mechanistic links to TRPV1.

2) We suggest using the term "sensitized" instead of "pre-sensitized" for K169A and related mutants, as their phenotype coincides with that of sensitized channels. We find the term pre-sensitized can be confounding.

We agree with the reviewers that the term “pre-sensitized” can be confusing. To avoid confusion, we have now changed the text so that it only refers to “sensitized” channels.

3) "due to impaired hysteresis" (subsection “Conformational changes of the distal CTD lead to rearrangements in the cytoplasmic inter subunit interface”) – may be a bit awkward. We suggest using "impaired sensitization" instead, as hysteresis is the phenomenological consequence of sensitization.

We see the reviewers’ point and have changed the references to “impaired hysteresis” to “impaired sensitization”.

4) Can the pi-helix in the S6 be unequivocally assigned at the resolution for each structure?

The S6 pi-helix is only observed in the TRPV3_K169A 2-APB_ structure (3.6Å), where the map is of sufficiently high quality around the S6 helix that the pi-helix can be assigned unambiguously. The density around S6 is shown in Figure 4—figure supplement 2 but we have now included a panel in Figure 4—figure supplement 2 to show a close-up of the pi-helical turn.

5) In the dose-response curves for the mutants with decreased activity, the apparent Kd values obtained are probably an overestimation, since the curves don't saturate. The authors should mention this, which in fact strengthens their conclusions.

We thank the reviewers for pointing this out: we have now included a sentence to point out that the dose-response curves to not saturate and that the Kd values are therefore likely to be higher.

6) The authors should use the experimental maps to provide a clear illustration of the conformational changes occurring at the switch such as those in Figure 2B,C, Figure 3C and Figure 4D, in which surface representations were used.

The electron density around the CTD is shown in Figure 2—figure supplement 2 (K169A), Figure 4—Figure supplement 2 (K169A 2-APB) and Figure 4—figure supplement 5 (K169A 2-APB, WT). However, no supplements were originally included to show the density around the HLH_CD_ and the loop of AR5 (to correspond to figures 3D and 4E). We have now included panels in Figure 4—figure supplement 4 to show the electron density around the switch in TRPV3_K169A_ and TRPV3_K169A 2-APB_.

7) The mouse TRPV3 channel mutation of residue N412 has been shown to have a large effect on sensitization (Liu and Qin, 2017). Is there any structural difference coming from the structures that could also explain the effects of this mutant?

This is an excellent question. We did discuss this study, but did not refer specifically to the N412 residue: “In line with our observations, a previous study has shown that manipulation of the length of the loop of HLH_CD_ affects sensitization properties of TRPV3 by increasing the coupling between the cytoplasmic and transmembrane domains [36].” The Liu and Qin study showed that the loop in HLH_CD_ is one amino acid shorter than that of TRPV1, and insertion of a valine residue into position 412 in this loop resulted in a sensitized TRPV3 channel. We believe that this insertion increases the coupling between the CD and the loop of AR5 and enforces the interactions that otherwise would be triggered by the coil-to-helix transition in the CTD.